# Dynamical modeling of the H3K27 epigenetic landscape in mouse embryonic stem cells

**Kapil Newar[1†], Amith Zafal Abdulla[2], Hossein Salari[2], Eric Fanchon[1], Daniel Jost[1,2]***

**1** Univ Grenoble Alpes, CNRS, TIMC laboratory, UMR 5525, Grenoble, France, **2** Laboratoire de Biologie et Modélisation de la Cellule, École Normale Supérieure de Lyon, CNRS, UMR 5239, Inserm, U1293, Université Claude Bernard Lyon 1, Lyon, France

† Deceased.
* daniel.jost@ens-lyon.fr

## Abstract

The Polycomb system via the methylation of the lysine 27 of histone H3 (H3K27) plays central roles in the silencing of many lineage-specific genes during development. Recent experimental evidence suggested that the recruitment of histone modifying enzymes like the Polycomb repressive complex 2 (PRC2) at specific sites and their spreading capacities from these sites are key to the establishment and maintenance of a proper epigenomic landscape around Polycomb-target genes. Here, to test whether such mechanisms, as a minimal set of qualitative rules, are quantitatively compatible with data, we developed a mathematical model that can predict the locus-specific distributions of H3K27 modifications based on previous biochemical knowledge. Within the biological context of mouse embryonic stem cells, our model showed quantitative agreement with experimental profiles of H3K27 acetylation and methylation around Polycomb-target genes in wild-type and mutants. In particular, we demonstrated the key role of the reader-writer module of PRC2 and of the competition between the binding of activating and repressing enzymes in shaping the H3K27 landscape around transcriptional start sites. The predicted dynamics of establishment and maintenance of the repressive trimethylated H3K27 state suggest a slow accumulation, in perfect agreement with experiments. Our approach represents a first step towards a quantitative description of PcG regulation in various cellular contexts and provides a generic framework to better characterize epigenetic regulation in normal or disease situations.

**Data Availability Statement:** The home-made Gillespie algorithm is implemented in Python 3.6 and is available at https://github.com/physical-biology-of-chromatin/PcG-mESC.

## Author summary

The regulation of gene expression in eucaryotes is in part regulated by specific biochemical modifications of chromatin, the so-called epigenetic marks. In particular, the Polycomb system deposits repressive marks that participate in the silencing of many genes during development. Recent experimental evidence suggested that the recruitment of specific enzymes (like PRC2) at dedicated genomic sites and their capacities to spread epigenetic marks from these sites are key to the functioning of the Polycomb repression. Here, we developed a mathematical model to test whether such mechanisms, as a minimal set of qualitative rules, are quantitatively compatible with data in mouse embryonic stem cells.

**Funding:** The research leading to these results was supported by the University Grenoble-Alpes via the SYMER program (which is funded by the French National Research Agency under the "Investissements d'Avenir" program ANR-15-IDEX-02). DJ acknowledges additional funding from the French National Research Agency [ANR-18-CE12-0006-03, ANR-18-CE45-0022-01]. EF acknowledges funding from ITMO Cancer in the framework of the French Plan Cancer (Systems Biology program). The funders had no role in study design, data collection and analysis, decision to publish, or preparation of the manuscript.

**Competing interests:** The authors have declared that no competing interests exist.

We showed that the model well predicts the epigenetic landscape around repressed genes as well as the kinetics of its establishment and maintenance. We demonstrated the key role of the reader-writer module of PRC2 and of the competition between the binding of activating and repressing enzymes in Polycomb regulation. Our approach represents a first step towards a predictive description of epigenetic regulation in various cellular contexts.

## Introduction

Cells sharing the same genetic information may have very different functions and phenotypes. The regulation of gene expression is central to control such cellular identity. In eukaryotes, a key layer of regulation lies in the modulation of the accessibility to DNA and in the recruitment of the molecules driving transcription to chromatin. In particular, biochemical modifications of DNA and of histone tails, the so-called epigenetic or epigenomic marks, are believed to be essential in controlling such modulation [1]. Each cell type is characterized by a distinct pattern of epigenetic marks along the genome with specific modifications associated with active or silent regions [2]. Such epigenetic information should be robust and maintained across DNA replication and cell divisions but may also need to be plastic and modified during development or to adapt to environmental cues [3]. A key question at the heart of epigenetics is thus to characterize the generic principles and mechanisms regulating the establishment, maintenance and conversion of the epigenomic marks.

Experimental studies suggested along the years that the regulation of these marks follows similar rules [4–6]: chromatin regulators like histone modifying enzymes (HMEs) are recruited at specific DNA sequences leading to the nucleation of an epigenetic signal that subsequently spread to form more or less extended domains along the genome. In particular, the spreading process was found to be driven by a variety of 'reader-writer' enzymes that can 'read' a given chromatin modification at a given locus and 'write' or 'remove' the same or another mark at other genomic positions [1,7].

To formalize such rules, several mathematical models investigating the generic regulation of histone marks have been developed [8–18]. In their simplest form, these models consider that the local chromatin state can switch between active and repressive marks [19,20]. They suggested that the reader-writer-eraser capacity of HMEs may generate positive feedback loops and cooperative effects in the system that are essential to provide stability to the local epigenetic state. Applications of such formalism, contextualized to specific marks at specific loci, have shown that it is fully consistent with many experimental observations [13,14,17,21–25]. However, quantitative comparisons with experiments are still rare in particular on how epigenetic marks organize around the nucleation sites, which may bring crucial information on the spreading and maintenance mechanisms [15,26]. In this work, we aim to provide a modeling framework able to quantitatively describe the genomic profiles of epigenetic marks in the context of the Polycomb system in mouse embryonic stem cells (mESCs).

The Polycomb regulatory system is found in many higher eukaryotes and has been shown to play a critical role during development in the silencing of lineage-specific genes [27]. It involves the methylation of the lysine 27 of histone H3 mainly via the coordinated action of two Polycomb-group (PcG) complexes, PRC1 and PRC2, tri-methylation of H3K27 (H3K27me3) being associated with gene repression. mESCs have been for years a model system to investigate the Polycomb system in mammals [28], as it is involved in the maintenance of the pluripotency of these cells [29]. Recently, many experimental studies in mESCs have

measured quantitatively the patterns of H3K27 modifications along the genome [6,26,30–34]. For example, genes targeted by PcG proteins, the so-called PcG-target genes, are characterized by high H3K27me3 levels around their transcriptional start sites (TSS) and intergenic regions are dominated by H3K27me2 representing more than 50% of all H3K27 modifications [6]. Perturbations of this H3K27 landscape when altering PRC1/2 have allowed to shed light on the functions of their molecular constituents in the establishment and maintenance of the epigenetic signal (Fig 1A) [6,30,31,35–37]. Briefly, the recruitment of a non-canonical PRC1 variant at CpG islands mediates locally the mono-ubiquitination of H2AK119 [35–40]. This localized signal in turn recruits PRC2 through the interactions of cofactors like JARID2 [6,41–43]. Around its core subunit Suz12, PRC2 contains the EZH1 or EZH2 catalytic subunit, both capable of methylating H3K27 and of nucleating the epigenetic signal. PRC2 includes also a 'reader' subunit, EED, that allosterically boosts the activity of the 'writer' EZH2 in presence of H3K27me3 residues [44] and allows the long-range spreading of the signal around the nucleation site [6]. Canonical PRC1 may then bind to H3K27me3-tagged regions leading to the local compaction of chromatin and repression of gene expression [35–37,45]. The silencing action of PcG proteins is antagonized by Trithorax-group proteins like MLL2 that recruit demethylases like UTX/JMJD3 and acetyltransferases like p300/CBP [46] mediating, respectively, the removal of the methyl groups from methylated H3K27 residues [47] or the addition of an acetyl group to unmarked H3K27 residues [48,49], which is crucial for transcriptional activation.

All this suggests that the recruitment of HMEs at specific sites and their local and long-range spreading activities from these sites are designing the epigenetic H3K27 landscape in mESCs. While these different mechanisms were discovered and characterized in a heterogeneous set of *in vivo* and *in vitro* experimental assays, it is unclear whether, all together, they provide a complete set of processes that can quantitatively describe *in vivo* experimental epigenetic profiles.

Here, to address this question, we turn this experimentally-derived knowledge into a quantitative mechanistic model. Building on previous generic mathematical models of epigenetic regulation (see above), we contextualize our framework to precisely account for the occupancy of key HMEs and for the major processes described above. Using this integrated model, we investigate how the 'spreading' tug-of-war between the repressive (H3K27me3) and active (H3K27ac) marks is tied to the properties and locations of HMEs around TSS. In particular, we show that the model predictions are in quantitative agreement with Chip-Seq H3K27 profiles measured around PcG-target, active or bivalent genes in wild-type (WT) and perturbed conditions and with Stable Isotope Labeling by Amino-acids Cell culture (SILAC) experiments monitoring the maintenance and spreading dynamics of H3K27 methylation. We demonstrate the central role of the reader-writer module of PRC2 and of the competition between activating and repressing mechanisms in shaping the relative levels of mono-, bi- and trimethylation around TSS. Finally, we conclude and discuss the perspectives and limitations of our approach.

## Results

### A dynamical model for the regulation of H3K27 epigenetic modifications in mESC

Based on previous experimental findings, we consider that the *de novo* establishment and maintenance of H3K27 modifications in mESCs is mainly carried out by the stable recruitment of HMEs to their cognate DNA recruitment sites and that the patterns of epigenetic marks

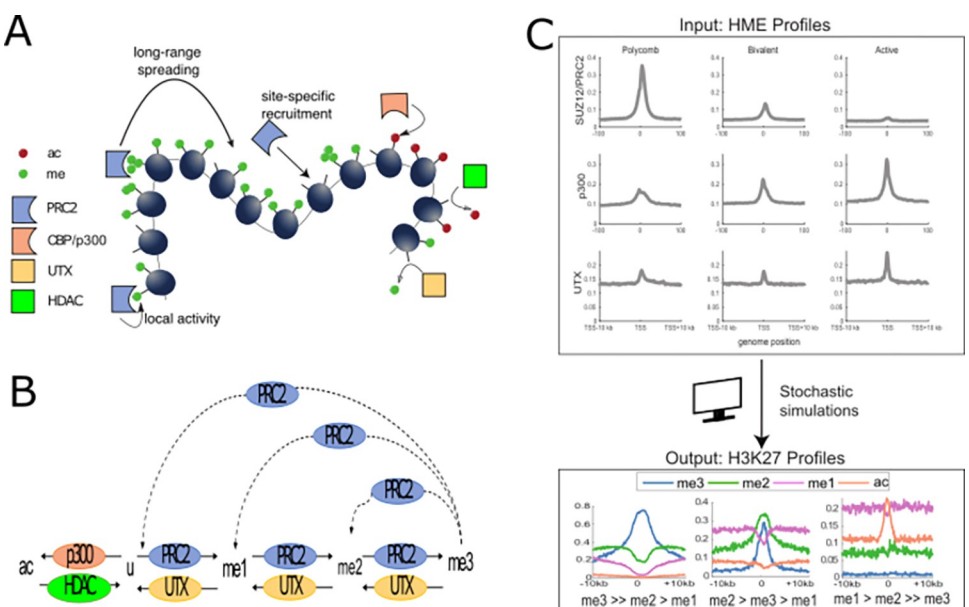

**Fig 1. Model for the regulation of H3K27 modifications in mESCs.** (A) Scheme of the different histone modifying enzymes (HMEs) involved in the regulation of H3K27 modifications, and of their actions. PRC2 complexes are recruited site-specifically and methylate H3K27 either locally, on site or at long-range if they are bound to trimethylated histones. CBP/p300 acetylates the H3K27 residues, UTX/JMJD3 and HDACs remove the methyl and acetyl groups from H3K27, respectively. (B) Multi-state dynamics of the H3K27 modifications: unmodified histones (u) are methylated to me1, me2 and me3 or acetylated to ac by the action of PRC2 (local and long-range) and p300 (local), respectively. Demethylation is conditional to the local UTX occupancy while deacetylation by HDACs is considered uniform. The long-range spreading of methylation mediated by PRC2-H3K27me3 allosteric activation is shown as dashed lines. Histone turnover and DNA replication are not shown here for clarity. (C) Summary of the computational framework with its inputs and outputs. The model takes HME profiles as inputs and makes predictions (based on the model depicted in (A,B)) on the probabilities to find a given mark at a given position and on methylation valencies, defined as the relative ordering of H3K27me levels around the TSS region.

around genes result from a complex network of local and long-range spreading or erasing mechanisms mediated by these HMEs [6] (Fig 1A).

To test this hypothesis quantitatively, we simulated the stochastic dynamics of H3K27 modifications in a 20kbp-region around the TSS of a gene. This region, made of 100 nucleosomes, is modeled as an array of 200 histones where we assume that each nucleosome (~200bp) is made of two consecutive independent H3 histones (each covering ~100 bp). The H3K27 status of each histone can fluctuate among five states (Fig 1B): unmodified (u), acetylated (ac), single-methylated (me1), double-methylated (me2) and tri-methylated (me3). Our model assumes that the dynamics of individual histones is mainly driven by the sequential addition or removal of acetyl and methyl groups by HMEs and by histone turnover [14,50–52]. Below, we describe the three main features of this model (Fig 1A and 1B). Their mathematical translation into reaction rates that control the stochastic transitions between histone states can be found in the Materials and Methods section.

**Addition and removal of the methyl groups by PRC2 and UTX.** Methylation of H3K27 is catalyzed by the PRC2 complex [53]. In mammals, PRC2 is predominantly recruited at CpG islands by several cofactors including JARID2 [6,41] or PRC1-mediated H2AK119 mono-ubiquitination [35–40]. Its methyltransferase activity is carried out by the subunits EZH1 or EZH2 [31]. While the EZH1 activity remains largely local, interactions between the PRC2 subunit EED and H3K27me3-marked histones at the core recruitment region may allosterically boost the EZH2 activity which is then allowed, via a reader-writer mechanism, to spread

methylation at long-range, outside the PRC2 cognate binding sites [6,30,44]. To account for this dual activity, we assumed that the methylation propensities at a given histone position are composed (i) by a local, on-site, nucleation term (of rate $k_{me_x}$ with $x \in \{1,2,3\}$) proportional to the PRC2 occupancy at this position; and (ii) by a long-range term (of rate $\epsilon_{me_x}$) accounting for the spreading capacity in 3D of distant PRC2 complexes bound to H3K27me3 histones at other positions that may spatially contact the locus by DNA looping [6]. As $k_{me_x}$ and $\epsilon_{me_x}$ rates reflect the catalytic activity of the same complex (PRC2), we further considered that the ratios $\epsilon_{me_x}/k_{me_x}$ ($x \in \{1, 2, 3\}$), which characterizes the fold-change in effective activity of the allosterically-boosted, long-range spreading vs local nucleation, is state-independent, ie $\epsilon_{me_x}/k_{me_x} = R$ for all $x$ (see Materials and Methods).

H3K27 methyl groups can be actively removed by the demethylase UTX with no evidence suggesting that UTX "spreads" its activity at long-range [47]. Therefore, we modeled the demethylation propensities as being local and simply proportional to the local UTX density with a rate $\gamma_{me}$ that, to simplify, we assumed to be independent of the methylation status.

**Addition and removal of H3K27 acetylation.** Acetylation of H3K27 is mediated by several acetyltransferases such as p300 or CBP recruited by transcription factors [46,48,49]. For p300-mediated acetylation, there is evidence suggesting that the bromodomain of p300 may trigger a reader-writer spreading process of acetylation [54], similar to the EZH2-mediated methylation. Such a mechanism would imply a long-range spreading of H3K27ac around the p300 binding sites. However, after analyzing H3K27ac and p300 ChIP-seq data around promoters, we found that p300 peaks are actually even slightly wider than the acetylation peaks (Fig 2B and 2C). Furthermore, while the inhibition of bromodomain enzymatic activity of p300 results in major loss of acetylation at enhancers, it only leads to minor changes at promoters [46]. Since we aimed to describe acetylation at promoters, we neglected the bromodomain interplay and we simply assumed an on-site enzymatic activity with the acetylation propensity being proportional to the p300 occupancy with a rate $k_{ac}$.

In general, deacetylation kinetics is fast and the half-lives of acetylated histone residues have been measured at many sites [55]. Therefore, we modeled the action of histone deacetylases (HDACs) as a uniform rate of 0.6 event per hour [56] acting on H3K27ac histones.

**Histone turnover and DNA replication.** In addition to the previous reactions that involve specific enzymes, the local state may be affected by histone turnover [58,60]. We assumed that this process leads to the replacement of the current histone state by a 'naive', unmodified (u) histone with a rate of 0.03 event per hour as measured consistently by two different studies [58,61].

DNA replication is also a major perturbative event for the epigenome as the 'mother' epigenetic information is diluted among the two sister chromatids [62]. Since mother histones are symmetrically redistributed [63,64], we modeled replication as specific periodic events, occurring every 13.5 hour (the median cell cycle length in mESC [59]), where half of the histone states are randomly lost and replaced by a 'u' state.

All together, these three main features drive the dynamic transitions between the different states of H3K27. This epigenetic model takes as inputs the binding profiles of HMEs like PRC2, p300 or UTX (Figs 1C and S1) and, for a given set of (de)methylation and (de)acetylation rates (Table 1), makes predictions about the corresponding profiles of H3K27ac/u/me1/me2/me3 modifications based on the simulations of many single-cell stochastic trajectories of small regions of the epigenome (see Materials and Methods).

In this paper, we focused on three gene categories in mouse ES cells grown in 2i medium for which lots of data are available (see Materials and Methods and Table 2): (i) Polycomb-

target genes are repressed and exhibit high H3K27me3 levels, low levels of H3K27me1 and me2 and the quasi-absence of H3K27ac; (ii) Active genes are enriched with H3K27me1 and H3K27ac marks; (iii) Bivalent genes, associated with lowly expressed or poised regions and characterized by the co-occurrence of active (H3K4me3) and repressive marks [65], exhibit intermediate levels of H3K27me2 and me3. Each of these categories, inferred from the statistical analysis of several transcriptomic and epigenomic information [26], are characterized by a distinct methylation valency—defined as the qualitative relative ordering of H3K27me levels—around the TSS (Fig 1C): me3>>me2>me1 for PcG-target genes, me1>me2>>me3 for active genes and me2>me3>me1 for the bivalent category.

## The model recapitulates the epigenetic landscape of Polycomb-target genes under various conditions

We first asked if our working hypotheses and the corresponding mathematical model are consistent with the average epigenetic landscape observed around PcG-target genes [26]. For that, we designed a multi-step inference strategy (Fig 2, S2 Fig) in order to fix, from available experimental data, the remaining free parameters of the model, namely the methylation nucleation rates ($k_{me1/me2/me3}$), the methylation spreading rates ($\epsilon_{me1/me2/me3}$), the demethylation ($\gamma_{me}$)

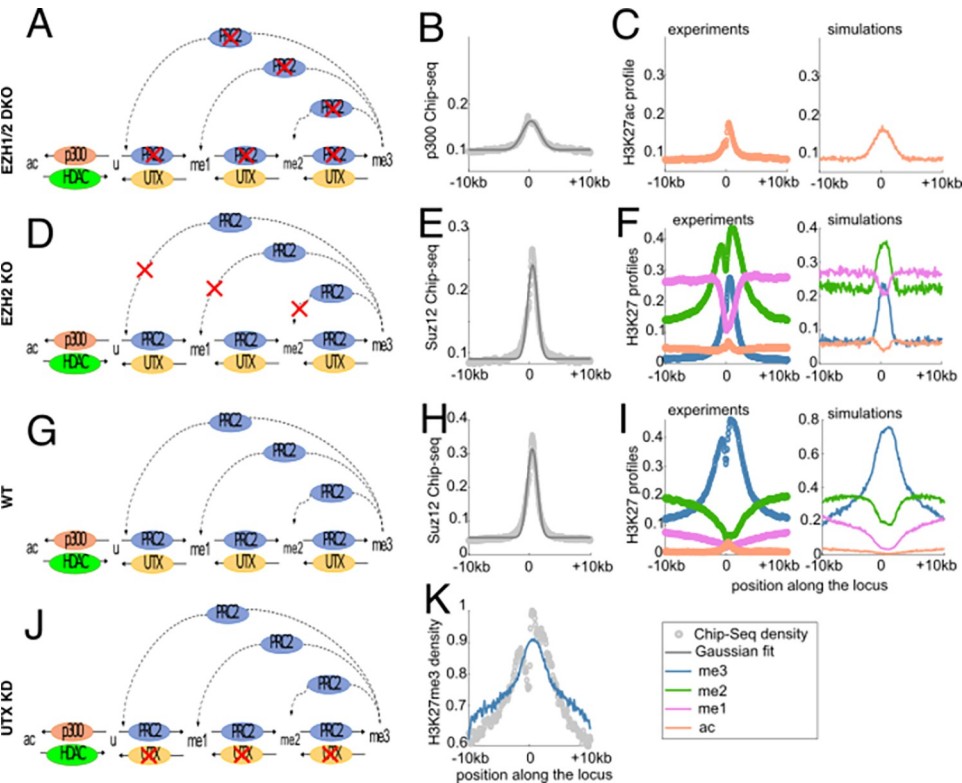

**Fig 2. Fitting the model with various perturbation experiments around PcG-target genes.** All profiles in the figure are around the TSS (position 0) of PcG-target genes. (A,D,G,J) Schematic representation of the epigenetic models used to simulate EZH1/2 DKO, EZH2KO, WT and UTX KD cases. (B) Average p300 occupancy in WT. (C) Fit (right) of the average experimental H3K27ac profile in EZH1/2 DKO cells (left). (E,H) Average SUZ12 occupancy for the EZH2KO (E) and WT (H) cases. (F,I) Experimental (left) and simulated (right) profiles of H3K27 marks for the EZH2KO (F) and WT (I) cases. (K) Fit of the average H3K27me3 profile for the UTX KD condition. In (B,C,E,F,H,I, K), circles correspond to normalized Chip-Seq profiles (gray for HMEs, colored for H3K27 marks), gray full lines to gaussian fits of the HME profiles and colored full lines to the predicted profiles of the epigenetic states.

and acetylation ($k_{ac}$) rates. In particular, we exploited data from wild-type and from perturbation experiments [30,33] where the activities of some HMEs have been modified.

**Latent acetylation of Polycomb domains in EZH1/2 double knockout.** In wild-type mESCs, the acetylated H3K27 histones sites are mostly spotted at the enhancers and promoters of active genes, overlapping with the genomic occupancy of both UTX and p300 [49,60]. Although p300 is present at the promoters of PcG-target genes (S1 Fig), there is almost zero H3K27 acetylation (orange circles in Fig 2I). However, on knocking out both methyltransferases EZH1and EZH2 [30], PcG genes become significantly acetylated (orange circles in Fig 2C) and a deregulation of gene expression is observed [30]. In this DKO situation where H3K27 methylation is absent, the epigenetic model reduces to a simple two-state model between $u$ and $ac$ states (Fig 2A). This allowed us to infer the acetylation rate $k_{ac} = 1.03 \ h^{-1}$ based on the p300 average occupancy around PcG promoters (Fig 2B) by fitting the corresponding average H3K27ac profiles (orange line in Fig 2C) (see Materials and Methods). Interestingly, our estimation of $k_{ac}$ is consistent with acetylation rates measured in human embryonic kidney cells for various residues after HDAC inhibition [66]. Since $k_{ac}$ is of the same order than the HDAC-mediated deacetylation rate ($\sim 0.6 h^{-1}$), it also suggests that acetylation levels result from a fast exchange dynamics of acetyl groups.

**Inference of methylation-related rates using EZH2 KO, WT and UTX KD profiles.** To infer the methylation-related parameters of the model, we designed an iterative scheme (S2 Fig) by sequentially using data from EZH2 KO, wild-type and UTX KD cells (see Materials and Methods section for details) for fixed ratios $r_{13} \equiv k_{me1}/k_{me3}$ and $r_{23} \equiv k_{me2}/k_{me3}$. (i) We started by initializing the nucleation rate $k_{me3}$ to an arbitrary value. (ii) Then, we took advantage of available data for EZH2 KO cells [30]. Indeed, in this strain, while the average PRC2 occupancy is maintained (Fig 2E), PRC2 loses its allosteric long-range spreading capacity (Fig 2D). This leads to a drastic reorganization of the methylation landscape (Fig 2F) with $me_2$ becoming the dominant methylation state in a 5 kbp-large portion surrounding the TSS (compared to the wild-type case on Fig 2I). By fitting this change of valency using a simplified version of the model with $\epsilon_{me1/me2/me3} = 0$ (no long-range spreading), we could infer the demethylation rate $\gamma_{me}$. (iii) Next, we reintroduced the spreading parameters (Fig 2G) and considered the wild-type profiles (Fig 2I) to fit the spreading-vs-nucleation ratio $R$ based on the methylation valency around promoters. (iv) After this step, there were no more free parameter in the model, however inferred $\gamma_{me}$ and $R$ values may depend on the initial guess made for $k_{me3}$ at stage (i). We thus used an independent dataset to validate our full set of parameters. For this, we compared the quantitative H3K27me3 profile given by MINUTE-ChIP experiment for UTX KD cells [33] (Fig 2K) to the model prediction where active demethylation by UTX has been inhibited ($\gamma_{me} = 0$) (Fig 2J). If not consistent, the same data allowed us to correct and optimize the $k_{me3}$ value, keeping all the other parameters fixed. By repeating steps (ii)-(iv) for this corrected value, we reevaluated $\gamma_{me}$, $R$ and possibly $k_{me3}$, until convergence (S2 and S3 Figs).

This overall inference strategy was applied to several values for $r_{13}$ and $r_{23}$ (S1 Table). Over all the tested cases, only one pair of ratios ($r_{13} = 3$, $r_{23} = 3$) leads to the convergence of the inference scheme (S3 and S4 Figs). Qualitatively, such ratio values are consistent with *in vitro* experiments on human EZH2 [50] showing a differential activity of PRC2 on H3K27u, me1 or me2 substrates with faster methylation rates towards me1 and me2 states than towards me3 ($r_{13} \geq r_{23} \geq 1$). Quantitatively, our estimation suggests that, *in vivo*, addition of the third methyl group (me2 to me3 transition) is a rate-limiting step for chromatin to acquire a H3K27me3—repressed—state, but at least 2 or 3 times less that initially observed *in vitro [50]*. The other optimal parameters are $k_{me3} = 0.81 \ h^{-1}$, $R = 0.85$ and $\gamma_{me} = 1.5 \ h^{-1}$ (Table 1). Interestingly, such close-to-one value for the spreading-vs-nucleation parameter $R$ suggests that the allosteric

**Table 1. Parameters of the epigenetic model.**

| Parameter | Description | Best value |
|---|---|---|
| $k_{me_1}$ | PRC2-mediated methylation rate ($u{\rightarrow}me_1$) | $3 \times k_{me_3}$ |
| $k_{me_2}$ | PRC2-mediated methylation rate ($me_1{\rightarrow}me_2$) | $3 \times k_{me_3}$ |
| $k_{me_3}$ | PRC2-mediated methylation rate ($me_2{\rightarrow}me_3$) | $0.81{\pm}0.1 \ h^{-1}$ |
| $R$ | spreading-vs-nucleation ratio ($\epsilon_{me_x}/k_{me_x}$) | $0.85{\pm}0.01$ |
| $\epsilon_{me_1}$ | EZH2 allosteric spreading rate ($u{\rightarrow}me_1$) | $R \times k_{me_1}$ |
| $\epsilon_{me_2}$ | EZH2 allosteric spreading rate ($me_1{\rightarrow}me_2$) | $R \times k_{me_2}$ |
| $\epsilon_{me_3}$ | EZH2 allosteric spreading rate ($me_2{\rightarrow}me_3$) | $R \times k_{me_3}$ |
| $\gamma_{me}$ | UTX-mediated demethylation rate | $1.5{\pm}0.05 \ h^{-1}$ |
| $k_{ac}$ | P300-mediated acetylation rate | $1.03 \ h^{-1}$ |
| $\gamma_{ac}$ | deacetylation rate | $0.6 \ h^{-1}$ taken from [57] |
| $\gamma_{turn}$ | histone turnover rate | $0.03 \ h^{-1}$ taken from [58] |
| $T$ | cell cycle length | $13.5 \ h$ taken from [59] |

boost of the EZH2 spreading efficiency mediated by H3K27me3 is of the order of 5- to 10-fold (see Materials and Methods), in very good agreement with *in vitro* experiments on human EZH2 [44,67].

While having an overall satisfying goodness of fit, the model still fails to capture some features observed in the ChIP-seq data. In EZH2 KO cells, the slowly-decreasing shape of H3K27me2 profile is not caught by the model (Fig 2F), suggesting the existence of an unknown putative spreading mechanism acting only on me2. In EZH2 KO and WT cells, the model predicts a decrease in H3K27ac levels at TSS while a slight gain is observed in the experiments (Figs 2F and 2I). In WT cells, the predicted profiles for methylation marks are ~1.5 fold stronger than the corresponding normalized ChIP-seq data (Fig 2I). Even if direct, absolute comparison between the magnitudes of model predictions and experiments should be done with great care (see Materials and Methods), this overestimation may translate a too strong $k_{me3}$ value.

**PRC2 spreading efficiency dictates the shapes, valencies and correlations of methylation profiles.** Our inference process illustrates how the epigenetic model and the underlying mechanistic hypotheses may consistently reproduce on the main lines the profiles of all H3K27 marks around PcG-target genes. Remarkably, while parts of the inference are based on qualitative fits of the average methylation valencies around the promoter, the model predicts quantitatively the inversion of valency occurring far from the promoter in EZH2KO and wild-type cells (Fig 2F and 2I respectively). We thus wondered what is the main mechanism driving such inversion. By varying the value of the spreading-vs-nucleation ratio $R$ while keeping HME profiles and other parameters as in the WT-case, simulations strongly suggest that valency around PcG genes is mainly driven by the long-range, allosteric spreading capacity of PRC2 (Fig 3A). When $R$ is very low (EZH2KO-like situation, $R{\sim}0$), me2 dominates at the nucleation sites (Fig 3A, TSS < 2.5 kbp) while me1 is predominant in the rest of the region (Fig 3A for distances to TSS> 2.5 kbp). Interestingly, in this low-$R$ regime, while the proportions of methylation states are overall limited, the model predicts that the levels of H3K27 acetylation are still low (less than 10%) everywhere and therefore we may expect that most of the PcG-target genes should remain inactive. This is in line with experimental observations showing that the massive transcriptional deregulation of PcG-target genes observed in EZH1/2 DKO cells is already rescued in EZH2 KO cells [32]. As $R$ increases, we progressively observed the emergence of H3K27me3 as the dominant state. The increase in me3 follows a switch-like,

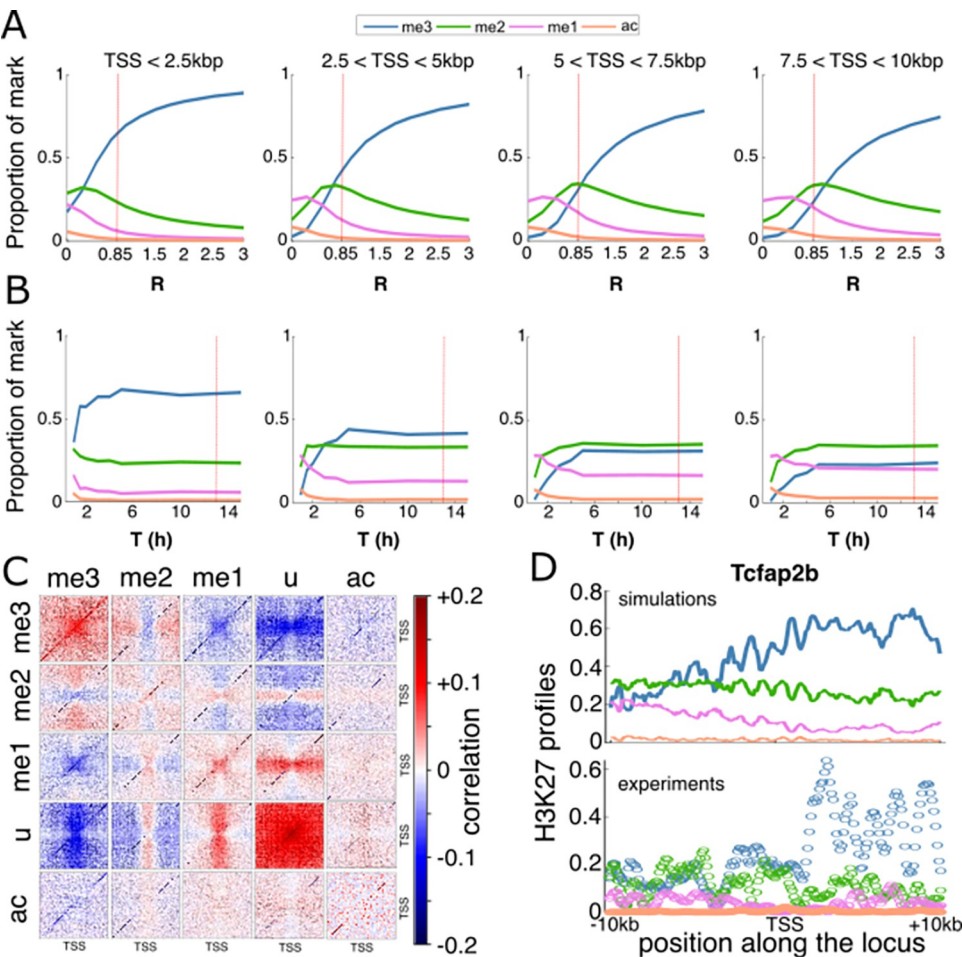

**Fig 3. Effect of spreading efficiency on PcG-target genes.** (A,B) Average predicted proportion of a given mark as a function of the spreading-to-nucleation ratio *R* (A) and of the cell cycle length *T* (B), all other parameters fixed to WT values (Table 1 and red dotted lines for *R* and *T*). Panels from left to right correspond to regions close or far from TSS. (C) Correlation between H3K27 states at different positions around TSS for the WT parameters. (D) Predicted (top) and experimental (down) WT profiles around the gene *Tcfap2b*.

sigmoidal, function as a function of *R* (blue lines in Fig 3A), signature of a phase-transition driven by the allosteric spreading. Strikingly, the inferred parameter for WT (*R* = 0.85, red dotted lines in Fig 3A) lies in the transition zone between the low and the high me3 regimes.

More generally, we performed a sensitivity analysis for all the model parameters (Figs 3A and 3B and S5). As expected, profiles around PcG-target genes are more sensitive to parameters related to the (de)methylation dynamics ($k_{me3}$, $\gamma_{me3}$) with $k_{me3}$ having a similar impact than *R*, and are more robust to variations of the other parameters, including histone turnover rate, around their WT values. Interestingly, the system becomes very sensitive to the cell cycle length for *T*≤6h (Fig 3B), by favoring the u-state by dilution during the replication process and thus by limiting the allosteric spreading. This suggests that regulation of the cell cycle length, that may vary from few to dozens of hours, as observed during differentiation and development [68] may participate in the global epigenetic regulation of gene expression [17,69].

As the spreading mechanism is constrained by the presence of H3K27me3 marks at the binding sites of PRC2, we expected to observe long-range correlations between the H3K27me3

level around the TSS where HMEs bind and the methylation state at more distal regions. More generally, to estimate the co-occurrence of H3K27 states at different positions, we computed from the simulated stochastic trajectories (S6 Fig) the correlations (see Materials and Methods) between the instantaneous local epigenomic state of any pairs of loci in the WT situation. Fig 3C illustrates the complex pattern of correlations existing between states and loci. As expected, positive correlations are observed between H3K27me3-tagged loci. Acetylated loci are weakly negatively (respectively positively) correlated with highly (resp. lowly) methylated states me2/3 (resp. me0/1). Due to the 'synchronized' dilution happening at replication, H3K27u-tagged loci are highly correlated. Other correlations between states translate the local competition between them and the long-range spreading capacity of H3K27me3 states at TSS. For example, me2 at (resp. out of) TSS is negatively (resp. positively) correlated with me3 everywhere. Indeed, me2 at TSS does not allow spreading methylation while me2 out of the nucleation region is the path towards me3 via spreading by me3 from TSS.

So far, we have parameterized and analyzed our model using the *average* experimental densities of HMEs around PcG-target genes as inputs and the corresponding *average* experimental profiles of H3K27ac/me1/me2/me3 modifications as outputs. Therefore, we wanted to test whether the same parameters are also viable for *individual* genes. Overall, plugging HME densities of *individual* genes into the simulations, we found that the individual profiles for each modification as well as the methylation valencies are well captured by the model (see examples in Fig 3D and S7 Fig). It is remarkable that we can still reproduce the specificity of each gene knowing that our parameterization has been based on an average signal that smoothed out these specificities.

## Parameter-free predictions of H3K27 modifications at active and bivalent genes

In our epigenetic model, the various methylation or acetylation patterns observed at different loci emerge from the differential binding of HMEs at these regions. We therefore asked if the shapes and valencies of H3K27 modifications observed around active and bivalent genes [26,30] may be predicted by the model using the same parameters as previously inferred at PcG genes but with the active and bivalent average HME profiles.

In EZH1/2 DKO cells, the model is able to predict quantitatively the acetylation profiles of bivalent and active genes using their respective corrected p300 profiles (Fig 4A–4D). In EZH2 KO cells, the model correctly predicts the methylation valencies for both active and bivalent genes but fails in capturing the acetylation level and the exact shapes of the methylation profiles (Fig 4E–4H). This suggests that the p300/UTX profiles that we took from WT as they were not available in the mutant strain, may be strongly perturbed in EZH2KO around non-PcG genes with a higher occupancy (as observed in EZH1/2 DKO cells).

In the WT case, active genes exhibit an inverse H3K27 methylation landscape compared to PcG domains with a valency me1 > me2 > me3 (Fig 4I). Interestingly, the average HME densities are also inverse compared to those around PcG-target genes: poor PRC2 binding and rich UTX/p300 occupancy (S1 Fig). Using these profiles as inputs, the model is able to correctly predict the average histone marks profiles including the observation that acetylation is higher than the methylation levels at the promoter (Fig 4J), the behavior around individual genes being also well captured (S8 Fig). At bivalent genes, H3K27me3 has the almost same peak density (average) as H3K27me2 around the promoter (± 2.5 kbp), both being higher than H3K27me1 and ac (Fig 4K). Our epigenetic model performs reasonably well in this region (Fig 4L) even if prediction for H3K27me3 is slightly lower than observed (see also S9 Fig). However, the model completely fails for more distal regions where ChIP-seq experiments show for example a flat profile for H3K27me2 that we don't capture. This suggests that other mechanisms not included in the model might play an important role at bivalent genes, like, for

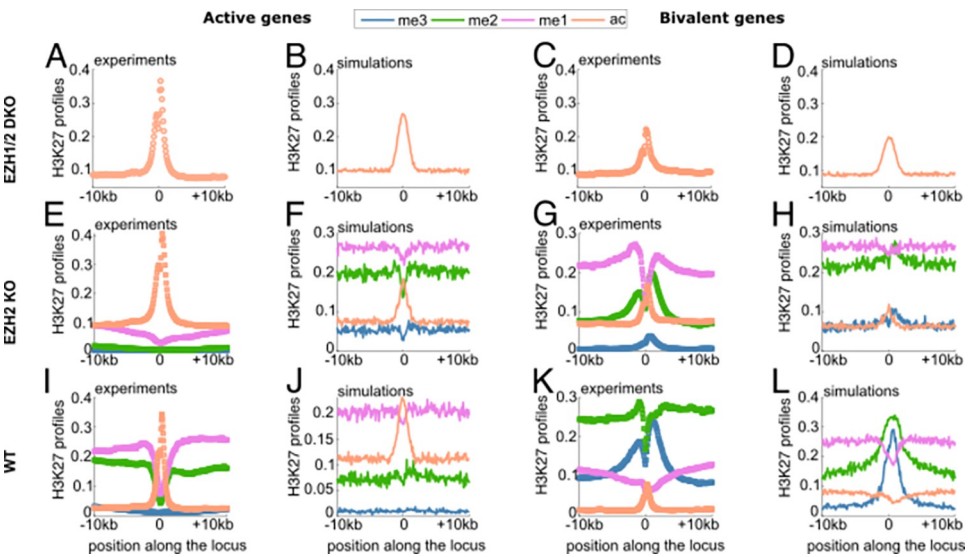

**Fig 4. Predictions of H3K27 modifications for active and bivalent genes.** Experimental (A,C,E,G,I,K) and simulated (B,D,F,H,J,L) profiles of H3K27 marks for the EZH1/2DKO (A-D), EZH2KO (E-H) and WT (I-L) cases around the TSS of active (A,B,E,F,I,J) and bivalent (C,D,G,H,K,L) genes. Symbols correspond to normalized Chip-Seq profiles and full lines to the predicted profiles of the epigenetic states.

example, the crosstalk with the regulatory machinery of other activating marks like H3K4me or H3K36me, also present at bivalent genes [33,34].

## Competition between activating and repressing factors shapes the local epigenomic landscape

More generally, we asked how the differential recruitment of HMEs around TSS may impact the local epigenetic landscape and subsequently gene regulation. To investigate this, we systematically computed the average profiles of H3K27 modifications as a function of the recruitment strengths of p300/UTX and PRC2 (see Materials and Methods). For each recruitment condition, we estimated the methylation valencies around TSS (S10 Fig) and found that the phase diagram of the system can be divided into 3 qualitative behaviors (Fig 5): (i) conditions with a PcG-target-like valency (me3>>me2>me1) observed for 'high' PRC2 and 'low' p300/UTX recruitments; (ii) situations with an active-like valency (me1>me2>>me3) for 'low' PRC2 and 'high' p300/UTX recruitments, and (iii) all the remaining, intermediate conditions, including bivalent-like valencies (me2>me3>me1). The approximate frontiers between these regions exhibit a stiffer dependency to the PRC2 occupancy, signature of the asymmetric tug-of-war between the acetylation by p300 and demethylation by UTX from one side and the methylation by PRC2 on the other side. The levels of p300/UTX at an average PcG-target gene in mESC (blue dot in Fig 5A) would need to be 3 times higher than the levels typically observed around active genes in mESC (orange dot in Fig 5A) to switch the gene into the active area for the same PRC2 level; while the level of PRC2 at an average active gene in mESC should be increased by 80% of the typical level found at PcG-target genes in mESC to move the gene into the PcG area. This suggests that activation of former PcG-target genes, during differentiation for example, at more reasonable levels of p300/UTX would require a concomitant decrease in PRC2 occupancy in parallel to the increase in p300/UTX recruitments. As repressors (PRC2) and activators (p300/UTX) binding motifs or recruitment signals are usually colocalized around TSS [70], a competition for their bindings to chromatin may naturally cause the inhibition of repressor occupancy while activator binding increases (or vice versa) [17].

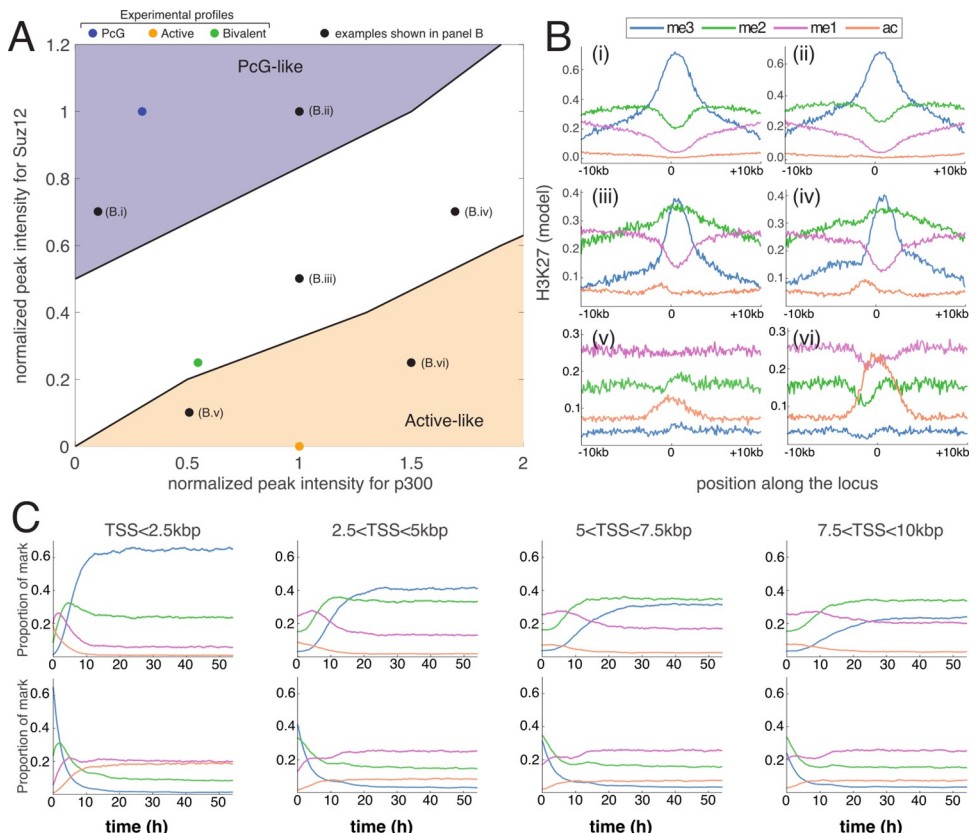

**Fig 5. Differential recruitment of HMEs at nucleation sites.** (A) Phase diagram of the model behavior obtained by varying the strengths of recruitment of p300 (x-axis) or Suz12/PRC2 (y-axis) around TSS for WT parameters. P300 (Suz12) peak intensities are normalized by the corresponding value at active (PcG-target) genes. The blue area represents situations where the methylation valency around TSS is PcG-target-like (me3>>me2>me1), the orange area to active-like conditions (me1>me2>>me3), the white zone to other cases including bivalent genes (me2~me3>me1). Colored dots give the positions of WT experimental profiles studied in Figs 2 and 4. Black dots are other special examples shown in panel (B). (B) Predicted epigenetic state profiles at different positions in the phase diagram (black dots in (A)). (C) Average predicted proportion of a given mark as a function of time after a switch, at *t = 0h*, from: (top) active-like (orange dot in (A)) to PcG-target-like (blue dot in (A)) HME profiles; and (down) from PcG-target-like (blue dot in (A)) to active-like (orange dot in (A)) HME profiles. Panels from left to right correspond to regions close or far from TSS as in Fig 3A.

## The model captures the maintenance and spreading dynamics of H3K27me3

Previously, we showed that the model well captures the average, 'static' epigenetic landscape around genes for a population of asynchronized cells at steady-state. We finally sought to use it to investigate the dynamics of regulation of H3K27 modifications.

We first interrogated how the epigenetic landscape is re-established after the strong, periodic perturbation of the system occurring every cell cycle at replication where half of the epigenetic information is lost. With the WT parameters inferred above, focusing on PcG-target genes, we tracked the dynamics of each H3K27 marks after replication for a population of synchronized cells in a periodic steady-state (Fig 6A). The me1 level rapidly increases up to ~4-fold (reached at ~*T*/7) and then slowly decays by 2-fold towards its pre-replication value. The me2 profile reaches almost its pre-replication value after ~*T*/3. The me3 level slowly grows along the whole cell cycle. This dynamics translates the gradual and slow re-establishment of H3K27me3 marks from the unmarked histones newly integrated at replication that are rapidly

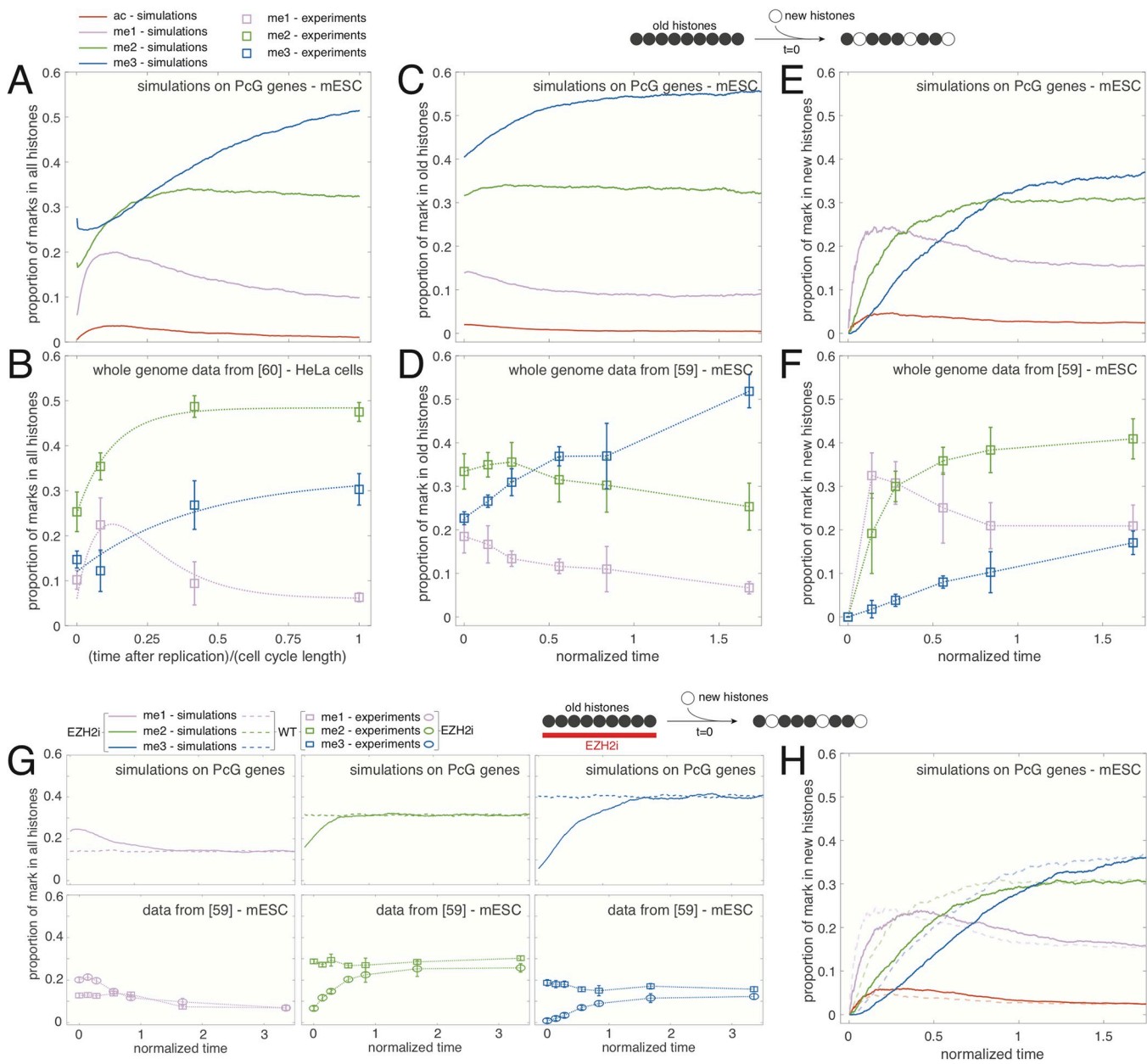

**Fig 6. Dynamics of H3K27 modifications.** (A,B) Intra-cell cycle dynamics for a population of synchronized cells as a function of the time after the last replication, predicted by the model for the WT parameters averaged around mESC PcG-target genes (A) or measured by SILAC experiments (B) on HeLa cells (data extracted from [62], see Materials and Methods). For each cell type, time is normalized by the corresponding cell cycle length ($T$ = 13.5h for mESC simulations, $T$ = 24h for HeLa cells). Dotted lines in (B) are a guide for the eye. (C-F) In WT conditions, we tracked from a given time (t = 0) the dynamics of marks in the pools of (E, F) newly incorporated (after turnover or replication) and (C,D) remaining (old) histones, for a population of unsynchronized cells. Predictions for the WT parameters averaged around mESC PcG-target genes are given in (C, E); SILAC experiments on mESC in (D,F) (extracted from [61], see Materials and Methods). Time is normalized by the effective histone decay time ($t_e$ = 12.7h for simulations, $t_e$ = 28.6h for experiments) (see Materials and Methods). (G, H) A population of unsynchronized cells is first evolved in presence (EZH2i) or absence (WT) of an EZH2 inhibitor. Then, at t = 0, if present, the inhibitor is washed out and the dynamics of marks in all histones (G) or in newly incorporated histones after t = 0 (H) is tracked. Predictions for the WT parameters averaged around mESC PcG-target genes are given in (G, top; H); SILAC experiments on mESC in (G, bottom) (extracted from [61], see Materials and Methods). Time is normalized as in (C-F).

methylated to me1, which represents a transient state towards higher methylation states. Even if me2 reaches a plateau suggesting that the 'me1 to me2' and 'me2 to me3' fluxes equilibrate, the system as a whole never reaches a steady-state during cell cycle due to the 'me2 to me3' rate-limiting step that occurs more slowly than the other transitions ($r_{13} = 3$, $r_{23} = 3$, see above). Remarkably, our predictions are in qualitative agreement with the cell cycle dynamics of whole-cell contents of H3K27me1/me2/me3 measured in human HeLa cells using SILAC (Fig 6B) [62]. The model even captures the small decrease in H3K27me3 level just after replication, that in our model can be interpreted by a significant demethylase activity not yet compensated by the methylation flux from the me2 state.

To better characterize the dynamics of newly integrated histones in the maintenance of a stable epigenetic landscape, we turned to a simpler system of unsynchronized cells where we tracked the time-evolution of the epigenetic state of unmarked histones incorporated in the region after a given time $t_0$ due to histone turnover or replication (scheme at the top of Fig 6C and 6E). For this pool of 'new' histones, we observed the same type of dynamics than along the cell cycle: me1 has a transient dynamics, me2 reaches a plateau and me3 grows very slowly (Fig 6E), confirming that the establishment of me3 marks on new histones extends over a long period (~2$T$). Consistently, the proportion of me3 in the pool of 'old' histones that were integrated before $t_0$ is still slowly increasing after $t_0$, while me2 levels remain almost constant and the me1 content slightly decreases (Fig 6C). Again, both predictions on old and new pools are qualitatively consistent with recent SILAC experiments performed on mESC (Fig 6D and 6F) [61], the recovery rate of H3K27me3 being even slower in the experiments than predicted.

Next, we analyzed the spreading dynamics of H3K27me3 around PcG-genes. We prepared the system as a population of unsynchronized cells evolving with a EZH2 inhibitor ($R = 0$, no spreading; WT values for other parameters) (scheme at the top of Fig 6G and 6H). Then, at a given time $t_0$, the inhibitor is washed out ($R = 0.85$, WT parameters) and we tracked the establishment of the epigenetic landscape after $t_0$ (full lines in Fig 6G, top). Due to the inhibition of EZH2, at t = $t_0$, the global level of me1 is higher than in normal WT condition (dashed lines in Fig 6G, top), while me2 and me3 are less present (see also Fig 2F and 2I). For t>$t_0$, we observed that the spreading of H3K27me3 and the recovery towards the WT state is slow and takes a few cell generations (~2$T$). These predictions are in perfect agreement with similar experiments on mESC studied with SILAC (Fig 6G, bottom) [61] or Chip-Seq [32] but also with EED-KO rescue experiments in mESC showing a recovery of the WT levels about 36 hours after the rescue [6]. After the release of the inhibition, we also followed the dynamics of newly incorporated histones (full lines in Fig 6H) and compared it to the dynamics of new histones but in WT conditions (Fig 6E and dashed lines in Fig 6H). Overall, we observed a shift of ~$T$/5 in the establishment of the epigenetic identity of the new histones compared to WT, consistently with SILAC experiments [61]. This delay is imputable to the weak density of H3K27me3 marks at the nucleation sites at $t_0$ that limits initially the long-range spreading by PRC2. This highlights the role of pre-existing H3K27me3 in controlling the dynamics of *de novo* methylation states thanks to the reader-writer capacity of PRC2.

Finally, we used the model to predict the establishment dynamics of new epigenetic landscapes after rapid and significant changes in the HME binding properties in an asynchronous cell population. We investigated differentiation-like situations where sets of active genes became repressed (Fig 5C, top) or vice-versa (Fig 5C, down). After a switch from active-like to PcG-target-like HME profiles, close to the TSS (<2.5kbp), we observed a successive "wave" of methylation with me1 being dominant, then me2 and finally me3. The establishment of the steady-state takes about one cell cycle and is mostly driven by the on-site action of HMEs. At larger genomic distances (>2.5kbp), establishment is slower (~ 2$T$) due to the time delay needed for the region close to the TSS to be trimethylated enough to allow allosteric long-

range spreading. For a switch from PcG-target-like to active-like profiles, a rapid, progressive wave of demethylation is visible around TSS. For larger distances, the kinetics is slowed down due to the allosteric spreading of the few PRC2 bound at a basal, background level at TSS which is still significant during a short time period just after the switch.

## Discussion and conclusion

In this work, we developed a model which accounts for the recruitment of HMEs at a domain of interest and then determines the histone modification levels as a consequence of a complex competition between the spreading and erasing capacities of these HMEs. In the light of rich quantitative data available from recent experiments, we picked up the case of H3K27 modifications in mESCs which allows, by modeling one residue, to investigate Polycomb-repressed, active and bivalent genes at the same time. By integrating key mechanistic details like the reader-writer capacity of PRC2 and experimental data of HME occupancy (Fig 1), this framework allowed us to analyze different conditions under one umbrella.

In particular, we inferred model parameters using data from WT and mutant conditions (EZH1/2 DKO, EZH2 KO, UTX KD) around PcG-target genes (Fig 2). We found that, to reach the repressive H3K27me3 state, the me2 to me3 transition was the limiting time step [50]. Our strategy also highlights the importance of looking at the full density profiles and methylation valencies around TSS (or nucleation sites) to efficiently estimate the spreading activity of PRC2. In particular, we estimated that the 'writing' efficiency of PRC2 is boosted by 5- to 10-fold when bound to H3K27me3 histones [44].

Our analysis suggested that such long-range, enhanced mechanism drives a transition between a low and high me3 regime and is essential for maintaining a proper H3K27me landscape (Fig 3A). The sigmoidal shape of this transition suggests that the epigenetic landscape could be sensitive to variations in the spreading efficiency [9]. While this may be advantageous for WT embryonic cells that are plastic and may need to differentiate rapidly following developmental cues, perturbations of this key allosteric capability may have deleterious impacts on gene expression. For example, gene deregulation in pediatric gliomas is associated with a loss of the EZH2 allosteric stimulation, mediated by the interactions between PRC2 and the oncohistone H3K27M or the oncoprotein EZHIP [71].

While the model was parameterized using average data around PcG-target genes, it was able to predict semi-quantitatively the H3K27 densities around individual PcG-target genes, but also around active and bivalent genes, by only plugging in the corresponding HMEs profiles (Fig 4). A systematic analysis of the role of HME recruitment (Fig 5A) allowed us to characterize the competition for epigenetic control between active and repressive factors. We showed that genes can be categorized as repressed or active depending on the levels of recruitment of activators and repressors. In particular, we observed that PRC2 binding at promoters, even at mild degrees, is essential to avoid spurious transcription by increasing the level of P300/UTX recruitment needed for transcriptional activation [14,34]. Activation of PcG-target genes should be necessarily accompanied by a significant decrease in PRC2 binding via, for example, the competition with activators for the binding at promoters [17,72]. Moreover, looking at the consistency between the predicted category of expression (silenced, active) and the observed one may be used to identify genes that are under the direct control of the H3K27 marks and associated HMEs.

Beyond the 'static', average description of H3K27 profiles around TSS, the model can be used to predict the dynamics of maintenance or establishment of the epigenetic landscape (Figs 5C and 6). In perfect agreement with experiments [61,62], we observed that the (re)formation of H3K27me3 domains *de novo* or after replication is a slow process in cycling mESCs.

This indicates that regulation of the cell cycle length during embryogenesis for example may also impact the stability and plasticity of the epigenetic landscape [17,69] (Fig 3B). We showed that the reader-writer capacity of PRC2 strongly influences such dynamics at PcG-target genes. Simply put, PRC2 which is recruited at promoters, first tri-methylates histones H3K27 at these nucleation sites, and then, thanks to allosteric activation, can tag more distal sites and spread methylation. The initial presence of H3K27me3 thus accelerates this process. This suggests that defects in partitioning of maternal H3K27me3 histones between the leading and lagging daughter strands [63,64] may generate asymmetries that may propagate to further generations as the H3K27me3 recovery dynamics is slow [73].

Previous generic models of epigenetic regulation [8–11,17] suggested mathematically that the maintenance of a robust and plastic epigenetic state may be associated with bistability that emerges from the self-propagation capacity of some epigenetic marks [8,9]. Here, we proposed, in line with recent experiments for the PcG system [6,32,74], that robustness is associated with the stable recruitment of HMEs at specific nucleation sites coupled to the long-range, allosterically-boosted spreading capacity of PRC2. As it is, our model cannot lead to mathematical bistability and does not support a self-sustainable, epigenetic memory: in our framework, PcG-target-like repression and bivalency are not representative of a stable state and of a rapidly-switching state, respectively, in a bistable regime [16], but rather correspond to a bimodal and a highly fluctuating state, respectively (S11 Fig). Interestingly, recent studies [32,75] showed that mESCs did not indeed have 'memory' as gene expression and H3K27 patterns are fully restored at their initial WT levels after the full removal of H3K27me3 (by removing PRC2 activity) followed by re-expression of PRC2.

Compared to the few other explicit models of PcG regulation in mammals [14,19,26], our framework also integrates a spreading process in competition with antagonistic erasing and activating processes, but the mechanistic nature of spreading differs. Chory et al [26] did not account for the allosteric enhancement of PRC2 and hypothesized that spreading from the nucleation site occurs via histone exchange between nearest-neighbor (NN) nucleosomes. Berry et al [14] and Holoch et al [75] considered the allosteric boost and that any H3K27me3 histone can spread methylation to its NN sites, allowing bistability to emerge. In both works, in addition to the self-propagation, 'reader-writer'-like capacity of PRC2, they also considered explicitly another feedback loop where transcription promotes demethylation (e.g., via an increase in histone turnover or via the recruitment of demethylase) and methylation inhibits transcription. In our framework, the effect of transcription/active states on the demethylation dynamics is also effectively integrated but as an open loop since the profile of the demethylase UTX is an input of the model and is highly correlated to p300 occupancy and to the gene transcriptional category. In terms of spreading, our formalism is closer to Erdel et al [15] that modeled H3K9me3 regulation in *S. pombe* from the methylation long-range activity of enzymes bound at a nucleation site.

In addition to the richness of available datasets, by modeling H3K27 regulation in mESCs, we were hoping that the high plasticity of these cells would allow us to explain their epigenetic landscape by simply focusing on main, primary mechanisms while neglecting *a priori* secondary effects. Our good description of the H3K27 profiles in different conditions around PcG-target genes and, to a lesser extent, around active and bivalent genes validated our approach. However, our predictions also contained several discrepancies, suggesting missing ingredients that may improve the description of H3K27 regulation in mESCs but also in more differentiated cell types.

A strong hypothesis that we made is that H3K27 profiles are only readouts of the HME occupancies via a complex network of interactions, but they do not feedback on the binding of HMEs or on the model parameters like UTX activity [47] or the histone turnover rate that may be impacted by transcription [14,26,76,77]. For example, the PRC2 'reader' subunit EED is

known to interact with H3K27me3 histones [44]. We assumed that such interaction was only relevant at nucleation sites to enhance EZH2 activity, but it may also lead to the recruitment of PRC2 at more distal sites and allow the self-propagation of the H3K27me3 mark [67]. We expect however such an effect to be weak in mESCs as the profile of H3K27me3 around TSS is much larger than the PRC2 binding density (Fig 2H and 2I) and a significant self-propagation would have led to profiles with more similar shapes. To account for such feedback, one would need to model explicitly the binding and unbinding of HMEs [18]. This would allow also to describe in more detail the role of PRC1 variants in nucleation and maintenance [35–37] or the competition for the binding of antagonistic HMEs around the same site [17].

To perform the parameter inference, we assumed that, in the different mutant strains, except the corresponding 'mutated' parameters that were set to zero (e.g., $R = 0$ in EZH2 KO), all the other parameters were not perturbed. However, changes in gene expression occurring in these cell lines [30] may have potentially modified the concentration or activity of regulatory proteins and thus may have impacted the other parameters.

Another simplification made in our work was to only model H3K27 modifications. While this might be sufficient to describe the regulation of PcG-target genes, a more accurate description of active and bivalent genes may require accounting for other 'active' modifications like the methylations of H3K4 and H3K36 by Trithorax-group proteins [16,61,78] that may interfere with H3K27me states [78]. Our model also does not explicitly account for the presence of histone variants at gene promoters like H2A.Z or H3.3 that may promote or impair respectively PcG regulation [79] and more generally for other mechanisms or modifications modulating the local chromatin structure and thus potentially interfering with HME binding or activity.

In our formalism, the long-range spreading of methylation results from 3D contacts between nucleation and distal loci. We have considered a simple, generic shape to describe genome folding in which such contacts were dependent only on the genomic distance ($P_{3D}(i,j)$ ~$1/|i-j|$). However, the model can be run for any $P_{i,j}$ matrices to account for specific locus-dependent organization, for example, to explain why H3K27me3 profiles around PcG-target genes in more compact regions are more extended (S12 Fig). Moreover, we also assumed that the 3D organization was fixed and not affected by the local epigenetic landscape. However, PRC1, that binds to H3K27me3, may form condensate [45,80–82], the so-called Polycomb bodies, and may subsequently impact the local 3D organization of the locus [83,84], an increased compaction that may in turn facilitate spreading [85–87]. Accounting for this positive feedback loop between long-range spreading and 3D chromosome organization may allow a better characterization of the role of genome folding in epigenetic regulation [88,89].

To conclude, the ideas and formalism developed here are general in nature and are adaptable to other cell types or epigenetic systems. In particular, it would be interesting to investigate the generality of the inferred parameters for more differentiated cells where H3K27me3 domains are usually more extended around PcG-target genes than in mESCs. It would allow to understand if these changes are solely due to differential HME binding, to modifications of rates like histone turnover [76,90] or to some unconsidered mechanisms as discussed above. More generally, our approach represents a first step towards a quantitative description of PcG regulation in various cellular contexts where 'secondary' effects may be integrated step-by-step to better estimate their importance in normal or disease contexts.

## Materials and methods

### ChIP-seq data analysis

We collected the raw Chip-seq data of various histone modifying enzymes (SUZ12, p300, UTX) and of H3K27-me3/me2/me1/ac marks from various sources as listed in Table 2.

**Table 2. List of Chip-Seq data used in this study.**

| Antibody | Cell line and perturbation | Experiment | Source |
|---|---|---|---|
| H3K27me3/me2/m1/ac | Wild type mESCs-2i | Chip-seq Spike-in | GSE116603 |
| SUZ12 | Wild type mESCs-2i | Chip-seq | |
| H3K27me3/me2/m1 | EZH2KO mESCs-2i | Chip-seq Spike-in | |
| SUZ12 | EZH2KO mESCs-2i | Chip-seq | |
| H3K27ac | EZH1/2 DKO mESCs-2i | Chip-seq Spike-in | |
| p300 | Wild type mESCs-Serum | Chip-seq | GSM2417169 |
| UTX | Wild type mESCs-Serum | Chip-seq | GSM2575693 |
| H3K27me3 | UTX/JMJD3 inhibited mESCs-2i | MINUTE-ChIP (calibration experiment) | GSM3595377 |

Corresponding fastq files were imported in the Galaxy environment [91] and mapped using *bowtie2* [92] to the mouse genome (mm9). For Chip-seq data with spike-in control, reads were mapped to the combined mouse + drosophila genomes (mm9+dm6). After removing the duplicates and sorting the bam files using *samtools [93]*, reads were normalized by the total number of mm9 mapped reads for normal Chip-seq and by the one of dm6 for Chip-seq with spike-in [94]. While utilizing all of these tools, we made sure that the same settings as in the original papers were used. To analyze the Chip-seq data, we first used *bamcoverage* from the deepTools 2.0 suite [95] to generate genomic profiles at a binning of 50bp. Then, to make quantitative comparisons between different histone modification levels, each profile was further normalized as in [26], using R, by dividing each bin value by the maximal value of the bin count over the genome, this maximum being estimated after removing outliers (bins outside the quantile range (0.1%,99.9%)). Normalized average profiles around TSS for PcG-target, bivalent and active genes were computed from the matrix files given by *computeMatrix* (deepTools 2.0) [95] for each gene list. The gene categories (PcG-target, bivalent, active) were directly taken from [26], in which Chory et al clustered genes using k-means based on CATCH-IT, RNA-seq, H3K4me3, H3K9me3, H3K27ac/me1/me2/me3 data.

The above method was used for normal ChIP-seq data except for the UTX/JMJD3 inhibited experiment (UTX-KD) obtained with MINUTE-ChIP [33]. In this case, the normalized bigwig file was directly sourced and fed to *computeMatrix*. Different from normal Chip-seq or Chip-seq spike-in experiment, the reads are normalized with total mapped reads of input to evaluate the input normalized read count (INRC). Then, the INRC at PcG-target genes is further scaled with the average peak density to approximate the actual density of H3K27me3 at PcG-target genes in UTX/JMJD3 inhibited mESCs.

## Description of the stochastic epigenetic model

**Mathematical formulation of the kinetic transition rates.** The dynamics of the epigenetic state is driven by kinetic rates accounting for the main features of the model described in the main text:

- Addition of one methyl group to histone *i* to reach state $me_x$ ($x \in \{1,2,3\}$) is governed by ($me_0 \equiv u$)

$$me_{x-1}(i) \rightarrow me_x(i) = k_{me_x}\psi_{suz12}(i) + \epsilon_{me_x}\sum_j P_{3D}(i,j)\psi_{suz12}(j)\delta_{j,me3} \qquad (1)$$

$k_{me_x}\psi_{suz12}(i)$ represents the local nucleation of the $me_x$ ($x \in \{1,2,3\}$) state with $k_{me_x}$ the corresponding rate and $\psi_{suz12}(i)$ the density of bound PRC2 at locus *i*. Practically, $\psi_{suz12}(i)$ is given

by the normalized Chip-Seq profile of Suz12 (see below). The right part in Eq 1 describes the long-range, allosteric spreading capacity of PRC2 with $\epsilon_{me_x}$ the corresponding enzymatic activity, $\delta_{j,me3} = 1$ if histone $j$ is trimethylated (= 0 otherwise) and $P_{3D}(i,j)$ the probability of contact between two histones $i$ and $j$. To simplify, we do not account for the locus-specificity of $P_{3D}(i,j)$ and assume that it only depends on the genomic distance $|i−j|$ between $i$ and $j$. Analysis of experimental Hi-C data showed that, in average, $P_{3D}(i,j){\sim}1/|i−j|^\lambda$, characteristic of the polymeric nature of chromatin [96]. In our simulations, we choose $\lambda = 1$ in accordance with Hi-C data in mESC [97].

- Removal of one methyl group to a $me_x$ histone ($x{\in}\{1,2,3\}$) at position $i$ is driven by:

$$me_x(i) \rightarrow me_{x-1}(i) = \gamma_{me}\psi_{UTX}(i) \tag{2}$$

with $\gamma_{me}$ the corresponding demethylation rates and $\psi_{UTX}(i)$ the density of bound UTX at locus $i$ that we extracted from the normalized ChIP-seq data.

- Addition of acetylation at histone $i$ follows the propensity:

$$u(i) \rightarrow ac(i) = k_{ac}\psi_{P300}(i) \tag{3}$$

with $k_{ac}$ the acetyltransferase activity of p300 and $\psi_{P300}(i)$ the density of bound p300 at locus $i$ that we extracted from the normalized ChIP-seq data.

- Removal of acetylation at histone $i$ is given by

$$ac(i) \rightarrow u(i) = \gamma_{ac} \tag{4}$$

with $\gamma_{ac} = 0.6\ h^{-1}$ the uniform deacetylation rate.

- Histone turnover leads to the loss of the current histone state replaced by a 'u' state:

$$X(i) \rightarrow u(i) = \gamma_{turn} \tag{5}$$

with $X{\in}\{ac, me_1, me_2, me_3\}$ and $\gamma_{turn} = 0.03\ h^{-1}$ the uniform turnover rate.

- DNA replication occurs every $T = 13.5\ h$. During this periodic event, the state of each histone can be lost with a probability ½ and replaced by a 'u' state.

**Relation between the nucleation and spreading rates.** In the epigenetic model (see above), a PRC2 complex bound to locus $i$ has a local (nucleation) activity with rates $k_{me_x}$ and may have, if $i$ is trimethylated, a long-range activity on any locus $j$ with rates $\epsilon_{me_x}P_{3D}(i,j)$ (Eq 1). Actually, this last term represents the allosterically boosted activity of PRC2 ($\epsilon_{me_x} \equiv F \times k_{me_x}$) times the probability $P_{3D}(i,j)$ for PRC2 in $i$ to contact $j$ in 3D with $F$ the fold-change of PRC2 activity due to allostery. Simple polymer physics arguments lead to $P_{3D}(i,j) \approx \sqrt{6/\pi}(a/d_{ij})^3$ with $a$ the typical 3D 'capture distance' of PRC2 and $d_{ij}{\approx}d_0|j−i|^{\lambda/3}$ the average distance between loci $i$ and $j$. Therefore $R \equiv \epsilon_{me_x}/k_{me_x} = F\sqrt{6/\pi}(a/d_0)^3$ is independent of $x$. Assuming that $a{\sim}10\ nm$ the typical size of PRC2 complex [98] and $d_{ij}{\sim}100$ nm for 10-kbp ($|j−i|{\sim}100$ histones) genomic distance [99] (i.e., $d_0{\sim}22\ nm$), $F{\sim}8{\times}R$.

**Simulations of the model.** For given parameters (Table 1) of the epigenetic model and for given profiles for Suz12, p300 and UTX (Fig 1B), the corresponding stochastic dynamics (S6 Fig) of the system was simulated using a home-made Gillespie algorithm [100] implemented in Python 3.6 (S1 Data) that is available at https://github.com/physical-biology-of-chromatin/PcG-mESC. All simulations were initialized to a fully unmodified state ('u' state for each histone) and run long enough to reach a periodic steady-state, independent of the initial configuration. For data in Figs 2, 3, 4 and 5, 32 independent trajectories per parameter set were simulated over 25 cell cycles to ensure the system has reached a periodic steady-state. Predicted average profiles of H3K27ac/u/me1/me2/me2 correspond to the probability for a given locus to carry a given mark in an asynchronous cell population (i.e., averaged over time and trajectories) at steady state. For data in Figs 5C and 6, 500 independent trajectories were simulated over 100h. In Fig 5C, the asynchronized population was first evolved for a given recruitment condition ($\alpha$;$\beta$) (see below) until steady state before switching to another condition at a given absolute time (t = 50h). In Fig 6A (cell cycle dynamics), average proportions along the cell cycle for each mark are given for a synchronized cell population (i.e., averaged over trajectories and over various cell cycles) that has reached a periodic steady-state. In Fig 6C–6E, average values correspond to a population of asynchronized cells (i.e., at a given *absolute* time, different trajectories may correspond to different *relative* times along the cell cycle) for which we tracked the replacement of histones by turnover or dilution from a given absolute time (t = 50h). In addition, for Fig 6G and 6H, the asynchronized population was first evolved in a EZH2i-like situation ($\epsilon_{me_x} = 0$) until steady state before being switched to a WT-like situation at a given absolute time (t = 50h).

**Profiles of HMEs used in the simulations.** To simulate the average epigenetic landscape around PcG-target, active or bivalent genes in 2i condition (Figs 2–4), we directly used the average normalized Chip-Seq profiles of p300, UTX and SUZ12 around TSS for each gene category (S1 Fig), as described above. Note that SUZ12 profiles have been well measured in 2i condition but p300 and UTX are taken from serum condition experiments (Table 2). However, we assumed that p300 and UTX profiles are also valid for the 2i condition for all gene categories.

For predictions around single genes (Fig 3D), the noisy normalized Chip-Seq profiles (50 bp-binning) were smoothed out with a moving average over a 300bp-long window.

To investigate the interplay between the recruitment strengths of p300/UTX and PRC2 (Fig 5), we first fitted the average HME profiles in WT (S1 Fig) by Gaussian-like functions: $\psi_m exp[-(i - i_0)^2/(2\sigma_0^2)] + \psi_b$ with $\psi_b$ the background level, $\psi_m$ the height of the binding peak from background, $i_0$ the position of the peak and $\sigma_0$ the typical width of the peak. Then, we predicted the average H3K27 proportions for hypothetical genes characterized by a Suz12/PRC2 profile $\psi_{Suz12}(i) = \alpha\, \psi_{m,Suz12}(PcG)exp[-(i - i_{0,Suz12})^2/(2\sigma_{0,Suz12}^2)] + \psi_{b,Suz12}$, a p300 profile $\psi_{p300}(i) = \beta\, \psi_{m,p300}(Act.)exp[-(i - i_{0,p300})^2/(2\sigma_{0,p300}^2)] + \psi_{b,p300}$ and a UTX profile $\psi_{UTX}(i) = \beta\, \psi_{m,UTX}(Act.)exp[-(i - i_{0,UTX})^2/(2\sigma_{0,UTX}^2)] + \psi_{b,UTX}$ where $\psi_{m,Suz12}(PcG)$ is the amplitude measured around PcG-target genes and $\psi_{m,p300/UTX}(Act.)$ around active genes, $\alpha$ and $\beta$ are two multiplicative factors allowing to vary the amplitudes of the HME profiles around TSS. Here, for simplicity, we assumed that p300 and UTX evolved with the same factor $\beta$. For example, WT PcG-target gene in 2i condition corresponds to ($\alpha$;$\beta$)≈(1;0.3) and active genes to ($\alpha$;$\beta$)≈(0;1).

## Parameter inference

To fix the remaining free parameters of the model, we develop a multi-step inference strategy based on the different perturbation experiments (EZH1/2DKO, EZH2KO, WT and UTXKD)

from [30,33]. To be self-consistent and potentially smooth out sequence-specific biases that may arise in ChIP-seq experiments, the fitting was carried out exclusively using the *average* experimental profiles around PcG-target genes.

Note that we normalized the ChIP-seq data in EZH1/2DKO, EZH2KO and WT (see above) with a method developed in [26] that allows quantitative *relative* comparisons between H3K27 profiles. However, we want to make it clear that these ChIP-seq profiles cannot simply be interpreted as *absolute* probabilities of being modified but rather as being proportional to such probabilities. That is why, most of our inference scheme is based on qualitative fits of the methylation valencies around TSS (see below). The only exception is the MINUTE-ChIP data in UTXKD (see above) that, after normalization, was shown to be representative of the true density of H3K27me3 histones over the genome [33] under the assumption that, in UTXKD, in absence of demethylase and demethylase-driven activation signals, PcG-target genes reach their maximum possible H3K27me3 levels.

**Acetylation rate.**   In EZH1/2 DKO cells, the epigenetic model becomes a two-state model between the 'ac' and 'u' states since methylation is not possible. In this simple case, the probability of being acetylated at position $i$ is given by

$$P_{ac}(i) = k_{ac}\psi_{p300}(i)/[k_{ac}\psi_{p300}(i) + \gamma_{ac} + \gamma_{turn} + log(2)/T] \tag{6}$$

The p300 ChIP-seq density $\psi_{p300}$ for EZH1/2 DKO cells was not available, so we used the wild type p300 occupancy with a correction factor. We first fitted the WT profile by a Gaussian-like function (full line in Fig 2B), $\psi_{p300}^{WT}(i) = \psi_m exp[-(i - i_0)^2/(2\sigma_0^2)] + \psi_b$, with $\psi_b =$ 0.1196 the background, non-specific binding level of p300, $\psi_m = 0.1666$ the specific maximal increase of binding at the peak, $i_0 = -0.5438$ the position of the peak and $\sigma_0 = 5.1875$ the typical width of the peak. Assuming a uniform difference in p300 occupancy between WT and DKO situations, we model the DKO p300 profile as $\psi_{p300}^{DKO}(i) = \psi_m exp[-(i - i_0)^2/(2\sigma_0^2)] + \alpha\psi_b$ with $\alpha<1$ the correction factor. Using this profile in Eq 6 and minimizing a chi-squared score between model predictions and experiments (Fig 2C), we can infer $k_{ac} = 1.03\ h^{-1}$ and $\alpha = 0.6$ (see also S13 Fig).

**(De)methylation rates.**   Inference of methylation-related parameters follows an iterative multi-steps strategy as described in the main text (see also S2 Fig).

1. Ratios $r_{13} \equiv k_{me1}/k_{me3}$ and $r_{23} \equiv k_{me2}/k_{me3}$ are fixed to arbitrary values.

2. The absolute nucleation rate $k_{me3}$ is initialized to a random value.

3. In our model, EZH2 KO cells correspond to an epigenetic system without spreading ($\epsilon_{me_x} = 0$). For various values of $\gamma_{me}$, we simulated this situation using the SUZ12 Chip-seq density measured in EZH2 KO by Lavarone et al [30]. We could not find the UTX/p300 occupancy of EZH2KO cells in literature, so wild type UTX/p300 occupancy was used. At this point, we do not see any indication that knockout of EZH2 will significantly alter the presence of UTX/p300 occupancy for PcG genes. As normalized Chip-seq densities are more pertinent in terms of relative comparison, we fixed the value of $\gamma_{me}$ that is qualitatively consistent with the methylation valency observed around the promoter of PcG-target genes in EZH2 KO cells, i.e., the profile of me2> profile of me3> profile of me1 at TSS (Fig 2F).

4. Using wild-type cells data, we then fixed the spreading rates $\epsilon_{me_x}$, or more exactly the ratio $R = \epsilon_{me_x}/k_{me_x}$. Again, estimation of $R$ was based on capturing the methylation valency at PcG-target genes rather than absolute Chip-Seq density, such that H3K27me3 is prevalent in the large regions around TSS's and eventually overtaken by H3K27me2 at ~5 (±0.5) kb from TSS (Fig 2I).

5. At this point, all parameters have been fixed ($k_{me1/me2/me3}$) or inferred ($\gamma_{me}$, $R$). We used MINUTE-ChIP experiments for UTX inhibited cells to validate these parameters. The corresponding H3K27me3 experimental profile has been calibrated such that it quantitatively corresponds to the probability that H3K27 is trimethylated [33] (see above) and can therefore be directly compared to model predictions. We simulated these cells with $\gamma_{me} = 0$ and neglecting the acetylation pathway considering that UTX is a stimulant for p300 recruitment [46]. By keeping $R$ to the inferred value in step 4 and $r_{13}$ and $r_{23}$ to the imposed values in step 1, we corrected the $k_{me3}$ value to minimize a chi-squared distance between predictions and experiments.

6. Steps 3 to 5 are repeated until convergence (S2 Fig) or failure of the fitting process when steps 3 & 4 cannot capture the experimental methylation valencies.

7. Steps 2 to 6 are repeated for different initial values of $k_{me3}$ (S3 Fig). We found that, in absence of failure, the strategy always converged to the same final parameter values.

8. Steps 1 to 7 are repeated for different values of $r_{13}$ and $r_{23}$ (S1 Table). We limited the scanning of these ratios to integer values between 1 and 9 with setting $r_{13} \geq r_{23}$ as suggested by the *in vitro* experiment [50].

## Correlations of the local epigenomic state

In Fig 3C, the correlation $C_{i,j}(x,y)$ between the state $x$ of locus $i$ and the state $y$ of locus $j$ ($x,y \in$ {$ac$, $u$, $me_1$, $me_2$, $me_3$}) is given by the Pearson correlation between the random variables $\delta_i(x)$ and $\delta_j(y)$ where $\delta_i(x) = 1$ if the H3K27 state of locus $i$ is $x$ (= 0 otherwise) in the current simulated configuration:

$$C_{i,j}(x,y) = (< \delta_i(x)\delta_j(y) > - < \delta_i(x) >< \delta_j(y) >)/$$
$$\sqrt{< \delta_i(x) >< \delta_j(y) > (1 - < \delta_i(x) >)(1 - < \delta_j(y) >)}$$

with $<.>$ the time and population average of the given random variable.

## Phase diagram driven by differential HMEs recruitment

The phase diagram given in Fig 5A was obtained by systematically varying $\alpha$ (from 0 to 1.4 every 0.2) and $\beta$ (from 0 to 2 every 0.2), the factors controlling the amplitudes of the PRC2 and p300/UTX profiles respectively (see above) and by computing the corresponding average H3K27 profiles. For each pair of values, we estimated the predicted mean proportion $\bar{P}_x$ of each H3K27 modification $x \in$ {ac,u,me1,me2,me3} in a ±2.5kbp window around TSS (S10 Fig). From this, we defined two regions characterized by a methylation valency qualitatively similar to the experimental valency observed in WT for PcG-target (me3>>me2>me1) and active (me1>me2>>me3) genes. More precisely, the PcG-target-like region was defined such that $\bar{P}_{me2} > \bar{P}_{me1}$ and $\bar{P}_{me3} > 1.5\,\bar{P}_{me2}$ and the active-like region by $\bar{P}_{me1} > \bar{P}_{me2}$ and $\bar{P}_{me2} > 1.5\,\bar{P}_{me3}$. The parameters that did not fall into these two regions form a third region that include methylation valencies compatible with bivalent genes.

## Data extraction from SILAC experiments

Model predictions on the dynamics of H3K27 modifications (Fig 6) are compared to experimental data obtained using the SILAC and mass spectrometry technologies that measure the global, genome-wide proportions of a given modification in different pools of histones [61,62].

Data in Fig 6B on the cell cycle dynamics of the different marks in all histones were obtained by averaging the proportions a given mark in the pools of old (light medium) and new (heavy medium) histones at different times after the release into S phase (which corresponds also to the moment of medium change) extracted from Figs 3E and S4C of [62]. Here, we arguably assumed that after replication and during one cell cycle, the pools of old and new are of similar sizes, as also done in Fig 3A of [62].

Data in Fig 6D and 6F on the dynamics of the marks in new and old histones were obtained from Figs 1D-F and S1A of [61]. The dynamics in the new histone pool were directly extracted from Fig 1F, left (Generation 3). The dynamics in the old histone pool were computed as the weighted average of the Generation 1 (extracted from Fig 1E, left) and Generation 2 (extracted from Fig 1D, left) data. The weight for each generation at a given time was taken proportional to the percentage of Generation 1 or 2 histones among all the histones (extracted from Fig S1A of [61]).

Data in Fig 6G (bottom) on the dynamics of the marks in all histones after the release of the EZH2 inhibitor were directly extracted from Fig S3C of [61].

In Fig 6A and 6B, to correct for differences in cell cycle lengths between simulations that are made on mESC and experiments made on HeLa cells, time after replication is normalized by the corresponding cell cycle length ($T$ = 13.5h for mESC simulations, $T$ = 24h for HeLa cells).

In Fig 6C-H, to correct for differences in global histone turnover rates between simulations and experiments, time is normalized by the corresponding effective histone decay time $t_e$ that captures the combined effect of direct histone turnover and of dilution after replication in a population of unsynchronized cells. $t_e$ is equal to the characteristic time of decay of the proportion of 'old' histones among all histones and can be obtained by fitted the corresponding curves by $exp(-t/t_e)$. For simulations, we estimated $t_e$ = 12.7 h. For experiments, we obtained $t_e$ = 28.6h by fitting the time-evolution of the sum of the proportions of Generation 1 and of Generation 2 extracted from Fig S1A of [61].

## Supporting information

**S1 Table. Tested combinations of $r_{13}$ and $r_{23}$.** Over all the tested cases, only $r_{13}$ = $r_{23}$ = 3 leads to a satisfying fit of the experimental profiles of H3K27 modifications around PcG-target genes (S3 Fig). Failure of the combination $r_{13}$ = $r_{23}$ = 4 is illustrated in S4 Fig.
(DOCX)

**S1 Data. Python code.** Jupyter notebooks with the python code used in the manuscript.
(ZIP)

**S1 Fig. HMEs profiles.** Average Chip-seq densities (normalized) of SUZ12, p300 and UTX of PcG-target, bivalent and active genes around the TSS in WT condition.
(TIF)

**S2 Fig. Iterative inference strategy.** For fixed values of $r_{12}$ and $r_{23}$, an initial guess for $k_{me}$ is used to initialize an iterative inference cycle where a parameter inferred at one step feeds (red arrows) the next inference step based on various datasets (black arrows): EZH2KO data to infer $\gamma_{me}$, WT for $R$ and UTXKD for $k_{me}$ (see main text).
(TIF)

**S3 Fig. Parameter inference for parameter $r_{13}$ = 3, $r_{23}$ = 3.** The steps for fixing $R$, $k_{me3}$ and $\gamma_{me}$ are illustrated. (A,D,G) H3K27 methylation are best fitted to EZH2 KO experimental profile by fixing $\gamma_{me}$ for a particular $k_{me3}$. (A,D,G) Then, H3K27 methylation are best fitted to WT

experimental profile by fixing $R$ for a fixed pair $k_{me3}$, $\gamma_{me}$. (C,F,I) Finally the fixed parameters $k_{me3}$, $\gamma_{me}$, $R$ are tested if the H3K27me3 profile fits experimental density.
(TIF)

**S4 Fig. Parameter inference for parameters $r_{13}$ = 4, $r_{23}$ = 4.** With these parameters, simulated H3K27 methylation profiles of EZH2 KO never capture the experimental methylation valency at promoters (Fig 2F of the main text). Top panel is for $k_{me3}$ = 0.9 $h^{-1}$ and explores $\gamma_{me}$ to find a suitable $k_{me3}$, $\gamma_{me}$ pair to qualitatively capture methylation valency of EZH2KO. Bottom panel explores $\gamma_{me}$ for $k_{me3}$ = 1.8 $h^{-1}$.
(TIF)

**S5 Fig. Systematic variation of model parameters.** Average predicted proportion of a given mark around PcG-target genes as a function of the different model parameters, all other parameters fixed to WT values (red dotted lines). Panels from left to right correspond to regions close or far from TSS. For $k_{me3}$, we also varied $k_{me1}$ and $k_{me2}$ by keeping $r_{13}$ and $r_{23}$ constant to WT values (see Table 1 of the main text).
(TIF)

**S6 Fig. Stochastic dynamics.** Kymograph representing a typical simulation trajectory at periodic steady-state around PcG-target genes with WT parameters obtained with the Gillespie algorithm. The local epigenetic state fluctuates stochastically following the kinetic rates given in the text.
(TIF)

**S7 Fig. Predictions of single PcG-target genes.** Predicting the H3K27 modification landscape at single genes in PcG-target domains. Profiles of gene *Tcfap2b* (left column), *Kcnq5* (middle column), and *Pcdh18* (right column). (First row) Input SUZ12 occupancies of three specific genes. (Second row) Chip-seq H3K27 methylation and acetylation for corresponding genes. (Third row) Simulated H3K27 modification profile of the respective genes.
(TIF)

**S8 Fig. Predictions of single active genes.** Predicting the H3K27 modification landscape at single genes of active domain. Profiles of gene *D11Wsu99e* (left), *Fam168b* (middle) and *Xpo5* (right). (First row) Input SUZ12 occupancies of three specific genes. (Second row) Chip-seq H3K27 methylation and acetylation for corresponding genes. (Third row) Simulated H3K27 modification profile of the respective genes.
(TIF)

**S9 Fig. Predictions of single bivalent genes.** Predicting the H3K27 modification landscape at single genes of bivalent domain. Profiles of gene *Scl6a1* (left), *Gm106* (middle) and *Xkr4* (right). (First row) Input SUZ12 occupancies of three specific genes. (Second row) Chip-seq H3K27 methylation and acetylation for corresponding genes. (Third row) Simulated H3K27 modification profile of the respective genes.
(TIF)

**S10 Fig. Competition between HMEs.** We varied the strengths of recruitment of p300 (x-axis) or Suz12/PRC2 (y-axis) around TSS for WT parameters. For each condition, we computed the average proportion of each H3K27 mark in a ±2.5kbp window around TSS. The corresponding stacked bar charts are given in the subplot (blue: me3, green: me2, magenta: me1, orange: ac, white: u). This allows us to define qualitatively two regions depending on the relative methylation valency: a PcG-target-like region (blue area) with me3>>me2>me1 and an

active-like region (orange area) with me1>me2>>me3.
(TIF)

**S11 Fig. Distribution of H3K27 states.** Probability distribution functions for the proportion of a H3K27 state inside the region TSS±2.5kbp in a population of unsynchronized cells for three values of HME recruitment strengths (Fig 5A of the main text), one in the Active-like region (($\alpha$;$\beta$)≈(0;1)), one with a bivalent-like inputs (($\alpha$;$\beta$)≈(0.4;0.5)) and one in the PcG-target-like region (($\alpha$;$\beta$)≈(1;0.3)).
(TIF)

**S12 Fig. Relation between compaction and H3K27me3 profiles.** For each PcG-target gene, we estimated a compaction score that translates the density of 3D contacts around this gene. More precisely, we took the Hi-C data of mESCs at 10kbp resolution from (Bonev et al, Cell, 171: 557–572.e24, 2017) that we distance-normalized (the Hi-C value of each bin *(i,j)* is normalized by the average contact frequency at genomic distance |*j-i*|) to obtain the so-called observed-over-expected contact matrix *OE*. For a gene *g* with a TSS at position $i_g$ along the genome, we define its compaction score as the log2 of the median value of the OE matrix in a ±100*kbp* window around the TSS:
$log_2[median\{OE((i_g - 100kbp) : (i_g + 100kbp); (i_g - 100kbp) : (i_g + 100kbp))\}]$. The distribution of compaction scores in the ensemble of PcG-target genes is given in panel (A). We divided this ensemble into three subgroups of the same size: Q1 with low compaction scores, Q2 with intermediate and Q3 with high scores (A). Panel (B) shows the average H3K27me3 profiles around TSS for each subgroup (computed as the other average H3K27 profiles in the main text). The more compact the gene is the more extended the profile is.
(TIF)

**S13 Fig. Fit of acetylation rate.** Experimental and fitted H3K27ac profile for different values of $\alpha$. The best fit $\alpha$ = 0.6 is picked for which $k_{ac}$ = 1.03 $h^{-1}$.
(TIF)

## Acknowledgments

This work is dedicated to the memory of our friend and colleague Kapil Newar, who passed away suddenly just before the acceptance of this manuscript. We acknowledge fruitful discussions with Cédric Vaillant, Guillermo Orsi, Sonja Prohaska and members of the SYMER consortium. We thank Emma Chory and Gerald Crabtree for sharing their ChIP-Seq normalization pipeline and the list of repressed, bivalent and active genes inferred in [26], as well as Simon Elsässer for useful insights on the MINUTE-ChIP method.

## Author Contributions

**Conceptualization:** Kapil Newar, Eric Fanchon, Daniel Jost.

**Data curation:** Kapil Newar, Hossein Salari.

**Formal analysis:** Kapil Newar, Amith Zafal Abdulla, Hossein Salari, Daniel Jost.

**Funding acquisition:** Eric Fanchon, Daniel Jost.

**Investigation:** Kapil Newar, Amith Zafal Abdulla, Hossein Salari, Daniel Jost.

**Methodology:** Kapil Newar, Eric Fanchon, Daniel Jost.

**Project administration:** Daniel Jost.

**Resources:** Kapil Newar.

**Software:** Kapil Newar.

**Supervision:** Eric Fanchon, Daniel Jost.

**Validation:** Kapil Newar, Eric Fanchon, Daniel Jost.

**Visualization:** Amith Zafal Abdulla, Hossein Salari, Daniel Jost.

**Writing – original draft:** Kapil Newar, Daniel Jost.

**Writing – review & editing:** Eric Fanchon, Daniel Jost.

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
