## [Decision Letter · Decision Letter 0]

10 Feb 2022

Dear Dr. Jost,

Thank you very much for submitting your manuscript "Dynamical modeling of the H3K27 epigenetic landscape in mouse embryonic stem cells" for consideration at PLOS Computational Biology.

As with all papers reviewed by the journal, your manuscript was reviewed by members of the editorial board and by several independent reviewers. In light of the reviews (below this email), we would like to invite the resubmission of a significantly-revised version that takes into account the reviewers' comments.

We cannot make any decision about publication until we have seen the revised manuscript and your response to the reviewers' comments. Your revised manuscript is also likely to be sent to reviewers for further evaluation.

Sincerely,

Ferhat Ay, Ph.D

Associate Editor

PLOS Computational Biology

Jian Ma

Deputy Editor

PLOS Computational Biology

Reviewer's Responses to Questions

**Comments to the Authors:**

**Reviewer #1**: The review is uploaded as an attachment.

**Reviewer #2: **This paper deals with an issue of current interest, namely, the mechanisms underlying the formation and sustainance of patterns of chromatin modifications or epigenetic landscape. Such landscapes are essential to the maintenance of cell identity and are known to be deteriorated in a number of situations, as for example, aging. Their model is based on a data-driven component, where the distribution of several epigenetic enzymes are extracted from Chip-seq data, and a modelling component, where several mechanisms for chromatin modifications are explored. Specifically, the authors provide an interesting exploration of the role of allostery in the PRC2 complex, which allows for non-local interactions between modified chromatin residues (H3K27me3). The methodology is sound and their results are interesting and in agreement with current experimental knowledge, so that I recommend this paper to be accepted for publication. However, I have some comments regarding issues that I think are in need of clarification.

1.- In Eq. (1), Description of the stochastic epigenetic model, the non-local part of the methylation rate is assumed to decrease as a power of the distance between loci measured in base pairs. Whilst this is a reasonable assumption, it is unclear whether it faithfully reflects the 3D structure of DNA which ultimately gives raise to the contacts between distant loci. Although later on the authors provide a rationale for this model in terms of polymer-physics argument, the authors should further clarify their assumption (for readers unfamiliar with polymer physics) and also comment on how the model could be modified in order to account for a network of contacts extracted form data

2.- Another issue regarding model formulation is that the authors consider only (positive) feedbacks between methylation marks. Other authors have consider other feedbacks between modifications and enzyme activity. Could the authors clarify this point?

**Reviewer #3: **The paper suggest a model for deducing relative strength of methylation reaction

associated to PRC2 by analysing how acetylation and me1, me2 and me3 marks are spatially distributed around a promoter. The overall assumption is that the output modification state

is a direct function of the input state, and thus that there is no significant feedback

for histones in me3 states recruiting PRC2, which again makes new me3.

Overall the work presents an interesting approach: By comparing experiments with and without

long range methylations from the assumed PRC2 binding sites the authors identify the me2me3 state as rate limiting. Also the overall estimates of rates seems reasonable.

Main objection is the moderate quality of the fit to data.

First of all, when reading figures, can the relative levels of different modifications

be compared in experiments? and does their absolute levels means anything?

Fig. 2 F: The model have constant levels of modifications outside TSS, while the experiments

for me2,me3 show longer range dependencies of distances (in fact this failure of reproducing

large distance trends go thorugh many of the figures,)

Fig. 2 I: levels boosted by factor 2 in model while y-axis in experiment unchanged.

Please comment on this and I would advice to be modest in claims

(is long range recruitment is too strong in model?)

Overall distance dependence of ac state appear to be opposite in model and experiment.

Fig. 4 F does not resemble panel E: The me1 in model are much

more localized. ac level in experiment are 4 fold enhanced at ESS while model only have 2 fold

Other panels in Fig. 4 show similar discrepancies, in various directions.

Overall the above does not inspire confidence for the reader, so please clarify.

From the pedagogical side, it is first quite late in the manuscript that one

understand that everything is just a readout of an assumed PCR2, profile.

It is mentioned early, but only briefly and Fig. 1B only gives a qualitative picture of something

added. How Much is added and where? Can P300 and UTX at all change anything away from

the TSS (since they seem to be a delta function there). How wide is the PRC2 profine addded?

(presumably that directly sets the width of the me2,me3 peak around TSS)

Overall Fig 1 does a poor job in explaining the model. What do one really learn from panel A

in that figure?

**Have the authors made all data and (if applicable) computational code underlying the findings in their manuscript fully available?**

Reviewer #1: None

Reviewer #2: Yes

Reviewer #3: Yes

PLOS authors have the option to publish the peer review history of their article (what does this mean?). If published, this will include your full peer review and any attached files.

Reviewer #1: No

Reviewer #2: No

Reviewer #3: No
---

## [Decision Letter · Decision Letter 1]

28 Jul 2022

Dear Dr. Jost,

We are pleased to inform you that your manuscript 'Dynamical modeling of the H3K27 epigenetic landscape in mouse embryonic stem cells' has been provisionally accepted for publication in PLOS Computational Biology.

Best regards,

Ferhat Ay, Ph.D

Associate Editor

PLOS Computational Biology

Jian Ma

Deputy Editor

PLOS Computational Biology

Reviewer's Responses to Questions

**Comments to the Authors:**

Reviewer #1: In this new version, Newar et al., the authors have carefully addressed every remark we had raised when examining the first version of their manuscript.

Both their answers and the modifications are satisfactory. In our opinion, this revised version of the manuscript is thus essentially ready for publication.

We noted a few small details that could easily be fixed

- In Fig1C, bottom panel, the color legend for the various methylation levels should be provided

- In section ‘Addition and removal of the methyl groups by PRC2 and UTX’ of the Results, there is a ‘vPRC1’ in the second sentence. We believe it is just ‘PRC1’.

- In section ‘Correlations of the local epigenomic state’ of the Math & Meth, there is an unnecessary repetition of the word ‘correlation’ in the first sentence.

Reviewer #3: I am happy with the revised manuscript and recommend publication without further reservations.

**Have the authors made all data and (if applicable) computational code underlying the findings in their manuscript fully available?**

Reviewer #1: Yes

Reviewer #3: Yes

PLOS authors have the option to publish the peer review history of their article (what does this mean?). If published, this will include your full peer review and any attached files.

Reviewer #1: No

Reviewer #3: **Yes: **Kim Sneppen

---

## [Editor Report · Acceptance letter]

18 Aug 2022

PCOMPBIOL-D-22-00005R1 

Dynamical modeling of the H3K27 epigenetic landscape in mouse embryonic stem cells

Dear Dr Jost,

I am pleased to inform you that your manuscript has been formally accepted for publication in PLOS Computational Biology. Your manuscript is now with our production department and you will be notified of the publication date in due course.

With kind regards,

Zsofia Freund
